# The Impact of Different Withering Process Conditions on the Bioactivity and Quality of Black Tea from Azorean *Camellia sinensis*

**Lisete Sousa Paiva [1,2,*], Ana Paula Dias [3], Massimo Francesco Marcone [4] and José António Bettencourt Baptista [1,2]**

[1] Department of Physics, Chemistry and Engineering (DCFQE), Faculty of Science and Technology, University of Azores, 9500-321 Ponta Delgada, Portugal; jose.ab.baptista@uac.pt

[2] Institute of Agricultural and Environmental Research and Technology (IITAA), University of Azores, 9700-042 Angra do Heroísmo, Portugal

[3] Gorreana Tea Plantation, Gorreana, 9625-304 Maia, Portugal; 2019112685@uac.pt

[4] Department of Food Science, University of Guelph, Guelph, ON N1G 2W1, Canada; mmarcone@uoguelph.ca

[*] Correspondence: lisete.s.paiva@uac.pt

**Abstract:** The objective of this study was to investigate the variability of natural bioactive compounds, such as catechin, theaflavin, total phenolic content (TPC), and total flavonoid content (TFC), of Azorean black tea (*Camellia sinensis* L., O. Kuntze) as well as its antioxidant activities according to different withering times. The TPC, TFC, free radical scavenging activity (FRSA), ferric reducing antioxidant power (FRAP), and ferrous-ion-chelating (FIC) activities were determined by colorimetric methods, and catechin and theaflavin contents were analyzed by high-pressure liquid chromatography (HPLC). The FRSA shows similar results for the withering range of 6 to 16 h (hours). For FRAP, the best results were observed at 16 h, and for FIC, the highest value was at 20 h. The TPC and TFC showed the highest value at 9 h and the lowest at 20 h. For the total theaflavins, the highest results were obtained after 12 h of withering, and the lowest values were obtained at 16 and 20 h. According to the different withering times, the highest value of total catechin levels was at 12 h, while the lowest value was observed at 20 h. Regarding caffeine content, all samples presented similar results, with the exception of the 12 h time point. In conclusion, the best withering times were observed in the range of 9 to 16 h, showing decreased values at 20 h, with the exception of FIC.

**Keywords:** tea antioxidants; total phenolic and flavonoid compounds; catechin profiles; theaflavin content; different withering process conditions

## 1. Introduction

Tea (*Camellia sinensis* L., O. Kuntze) is one of the most ancient as well as one of the most widely consumed non-alcoholic beverages, and its popularity is attributed to its sensory properties, stimulating effects, potential health benefits, and relatively low retail price. *C. sinensis* tea consumption has been proven to benefit human health, and *C. sinensis* polyphenol compounds, particularly the tea catechins, play important roles, displaying antioxidant properties [1] as well as anti-β-Amyloid [2], antidiabetic [3], antiallergy [4], anticancer [5,6], anti-inflammation [7], and antiviral effects [8]. Polyphenols are also considered to be a promising antiviral agent against multiple viruses, such as Ebola virus (EBOV) [9], influenza A (H1N1) [10], and even SARS-CoV-2 (COVID-19) [11,12]. Theaflavins (TFs) are another abundant class of tea polyphenols found in black tea that are responsible for the color and taste of tea infusions. The TF derivatives found in black tea are theaflavin (TF → EC + EGC), theaflavin-3-*O*-gallate (TF-3-G → EC + EGCG), theaflavin-3′-*O*-gallate (TF-3′-G → ECG + EGC), and theaflavin-3,3′-*O*-digallate (TF-3,3′-DG → ECG + EGCG), obtained from the combination of tea catechins which are generated as a result of polyphenol oxidase and/or peroxidase enzymatic activities.

Tea quality can be significantly impacted by many factors, such as leaf age, plant variety, plucking season, climate, genetic strain, and tea processing, as well as agronomic management and storage conditions [13,14]. However, one of the most widely consumed teas worldwide is black tea (75%), in which the catechins presented in the tea leaves are oxidized during the fermentation/oxidation process. Black tea processing, which starts from the harvesting of fresh tea leaves, consists of four sequential steps: withering, maceration/rolling, fermentation/oxidation, and finally drying. The withering stage, which occurs in the tea leaf from the time it is detached from the plant (plucked) to the time of maceration [15], has much to do with poor field tea leaf handling, losses during transit to the factory, and losses during the withering time inside of the factory. Typically, fresh leaves arriving at the factory have a moisture content above 75%, but this can also be as high as 84% on a wet basis. On the other hand, a maceration apparatus operates most efficiently in a green-leaf moisture range of 65–72% on a wet basis. According to Wilkie [16], up to 25% of the tea market value can be lost before the fresh green leaves arrive at the tea factory door. Therefore, to produce high-quality black tea, improving aroma, flavor, and other benefits, as well as the proper withering time, is deeply important [17]. Biochemical and physiological changes will occur, and consequently, the duration of withering plays an important role in these interactions and, therefore, in the tea quality [18,19]. The relative humidity, temperature, and time are the major factors affecting physical withering [20], and tea researchers advise that the withering process should be carried out below 38 °C when aiming to produce high-quality black tea [21]. There are mainly two types of withering: chemical and physical [19,22]. As the degree of chemical withering progresses, the permeability of the tea leaf cell membranes increases, which results in the leaf becoming flaccid. This type of withering refers to natural biochemical changes that occur inside the leaf, where complex chemical compounds are being broken down into simpler ones. In physical tea withering, shoots lose moisture, dropping from approximately 70–80% to 60–70% (wet basis), and the turgid leaf becomes flaccid. This process also leads to a higher sap concentration in the cells of the tea leaf [21]; consequently, the enzyme concentration increases, and the formation of high-molecular-weight compounds from low-molecular-weight subunits will occur as a result. On the other hand, catechin content decreases during withering [23], while theaflavins and their polymers, thearubigins [24], are formed, these being two tea components essential for the color, brightness, briskness, and strength of black tea. According to Omiadze et al. [24], withering time should be limited to 18 h, and according to Owuor and Orchard [25], withering time beyond 20 h leads to the deterioration of black tea quality. However, there is no specified withering duration, despite 14 to 18 h being generally accepted as the optimum range [21]. According to Baruah et al. [22], the quality of black tea depends on the chemical withering initiation, anaerobic, or catabolic phases, and during the withering process, a reduction in polyphenol oxidase (PPO) enzyme activity also affects the oxidative condensation of flavanols [26]. Withering temperature, as already mentioned, is another important factor for tea quality, and excessively high temperatures lead to leaf cell matrix destruction and, consequently, early uncontrolled fermentation-like reactions [27]. Moreover, black tea production at high temperatures leads to the initiation of unfavorable enzymatic reactions, producing undesirable amounts of theaflavins and thearubigins that are responsible for decreasing or increasing flavor, brightness, and sensory parameters [28]. Knowing the strong impact of withering on tea quality and the fact that, as far as we know, this is the first time a study has been performed to determine the effect of different withering conditions on the quality of Azorean *C. sinensis* black tea, the objective of this study was to investigate the variability of natural bioactive compounds, focusing particularly on catechins, theaflavin contents, total phenolic content, and total flavonoid content, as well as the antioxidant activities (FRSA, FRAP, and FIC), according to different withering times.

## 2. Materials and Methods

### 2.1. Chemicals and Reagents

Catechins, namely (−)-gallocatechin (GC, 98%—G6657), (+)-catechin (C, 98%—C1251), (−)-epicatechin (EC, 98%—E4018), (−)-epigallocatechin (EGC, 98%—E3768), (−)- epigallocatechin-3-gallate (EGCG, 95%—E4143), (−)-epicatechin-3-gallate (ECG, 98%—E3893), (−)-gallocatechin-3-gallate (GCG, 98%—G6782), (−)-catechin-gallate (CG, 98%—C0692); caffeine (CAF, 99%—C0750); gallic acid (98%—G7384); theaflavin (TF); theaflavin-3-*O*-gallate (TF-3-G); theaflavin-3′-*O*-gallate (TF-3′-G); theaflavin-3,3′-*O*-digallate (TF-3,3′-DG); ethylene-diaminetetraacetic disodium salt (EDTA); 2,2-diphenyl-1-picrylhydrazyl (DPPH); butylated hydroxytoluene (BHT); potassium ferricyanide; iron (II) chloride ($FeCl_2$); iron (III) chloride ($FeCl_3$); aluminum chloride ($AlCl_3$); ferrozine; trichloroacetic acid (TCA); Folin–Ciocalteu reagent (FCR); and rutin, were all obtained from Sigma-Aldrich (St. Louis, MO, USA). Sodium phosphate ($NaH_2PO_4$), sodium chloride (NaCl), sodium carbonate ($Na_2CO_3$), potassium acetate ($KCH_3CO_2$), and orthophosphoric acid were obtained from E. Merck (Darmstadt, Hessen, Germany). HPLC-grade acetonitrile (ACN) and methanol (MeOH) were purchased from Fluka Chemika (Steinheim, Switzerland). Chloroform and ethyl acetate, HPLC-grade, were obtained from Riedel-de Häen (Aktiengesellschaft, Seelze, Germany). Ultrapure glass-distilled water that was deionized with a Millipore Milli-Q purification system (Millipore, Bedford, MA, USA) was used throughout all the experiments.

### 2.2. Tea Sample Preparation

Collected tea leaves from Azorean *Camellia sinensis* (L.) Kuntze var. *sinensis*, containing the apical bud and the two youngest leaves, were provided by the Gorreana Tea Plantation (São Miguel Island, Azores, Portugal—37°49005.900 N 25°24008.200 W). Knowing the effect of sample preparation conditions (e.g., time and temperature of extraction, type of solvent, and solute:solvent ratio) on the tea extraction yield, the tea samples were prepared under the following conditions: tea leaves freshly plucked were indoor-withered at 25–30 °C for 6, 9, 12, 16, and 20 h; oxidized for 3 h; and dried in a heating chamber at 70 °C with rotating fan to keep the heat evenly distributed. The weight of the tea leaves was checked from time to time until a constant weight was reached. Then, the tea leaves were ground in a mortar to a particle size of 20–30 μm and refrigerated under an atmosphere of $N_2$ until further analysis.

### 2.3. Tea Extract Preparation

The *C. sinensis* aqueous extracts were prepared using 1 g of dried powder material in 20 mL of Milli-Q water under an atmosphere of $N_2$ to prevent oxidation. To mimic tea brewing, these solutions were heated at 70 °C in a water bath for 15 min to avoid the degradation of catechins that can occur at temperatures higher than 70 °C. The extraction process was repeated three times, under the same conditions, and the combined extracts were filtered, under vacuum, through a cellulose acetate membrane (porosity of 0.45 μm) to remove particulate matter. Then, they were dried on a rotary evaporator and lyophilized for further analysis.

### 2.4. Determination of the In Vitro Antioxidant Activity of Tea Extracts

To better characterize the antioxidant properties of tea natural bioactive compounds, obtained following the methodology described in Section 2.3, the tea extracts were evaluated using different in vitro antioxidant assays, such as DPPH-FRSA, FRAP, and FIC.

#### 2.4.1. Determination of DPPH Free Radical Scavenging Activity (FRSA)

The DPPH-FRSA assay, based on both electron transfer and hydrogen atom transfer reactions, is one of the most common assays for the determination of FRSA according to the method of Molyneux [29] with some modifications [30]. The FRSA of each tea extract, at various concentrations, was determined in triplicate by measuring its ability to scavenge DPPH, a stable free radical. An aliquot of 250 μL of each sample extract or BHT was added

to 500 μL of a 100 μM DPPH solution. BHT was used as the reference sample at the same concentration of the tea extracts, and a mixture without a sample or BHT was used as the control. After a post-incubation period of 30 min at room temperature in darkness, the Abs was measured at 517 nm. The FRSA was calculated as a percentage of DPPH discoloration using the following equation: $FRSA\ (\%) = (1 - Abs\ sample \div Abs\ control) \times 100$.

The results were expressed as an $EC_{50}$ value (μg/mL), which is defined as the sample concentration that can quench 50% of the DPPH free radicals. A lower $EC_{50}$ value is indicative of higher antioxidant activity.

### 2.4.2. Determination of Ferric Reducing Antioxidant Power (FRAP)

The total reducing power of each tea extract was determined according to the previously described methodology reported by Oyaizu [31] with slight modifications [30]. An aliquot of 0.4 mL of each extract sample was mixed with 0.4 mL of 200 mM phosphate buffer at pH 6.6 plus 0.4 mL of 1% potassium ferricyanide (*w/v*). This mixture was incubated at 50 °C for 20 min, and 0.4 mL of 10% TCA (*w/v*) was added to the mixture to stop the reaction. After that, the mixture was centrifuged at $4000 \times g$ for 10 min. The upper layer was separated into aliquots of 1 mL, and each one was diluted with 1 mL of deionized water plus 0.2 mL of 0.1% $FeCl_3$ (*w/v*). BHT was used as a reference at the same concentration of the extracts. An increase in the Abs values indicates an increased reducing power of the samples. The results were expressed as an $EC_{50}$ value (μg/mL), which is the concentration at which the Abs was 0.5 for reducing power, and were obtained by interpolation from linear regression analysis of concentration versus Abs at 700 nm against a blank.

### 2.4.3. Determination of Ferrous-Ion-Chelating (FIC) Activity

The ion-chelating ability of each tea extract was determined following the method of Wang et al. [32] with some modifications [30]. The chelating ability of each extract, at various concentrations, was determined by measuring the inhibition of the formation of the $Fe^{2+}$–ferrozine complex. An aliquot of 100 μL of each tea extract sample was mixed with 135 μL of methanol plus 5 μL 2 mM $FeCl_2$. The reaction was started by the addition of 10 μL of 5 mM ferrozine solution. After 10 min at room temperature, the Abs was measured at 562 nm. Methanol, instead of ferrozine solution, was used as a blank sample, which is required for error correction due to the unequal coloration of the sample solutions. Methanol, instead of a sample solution, was used as a control. Results were expressed as relative iron-chelating activity compared to the unchelated (without ferrozine) $Fe^{2+}$ reaction, and EDTA was used as the reference standard. The FIC activity was calculated as follows: $FIC\ activity\ (\%) = (A0 - (A1 - A2)) \div A0 \times 100$, where *A0* is the Abs of the control, *A1* is the Abs of the sample or standard, and *A2* is the Abs of the blank.

### 2.5. Determination of Total Phenolic and Total Flavonoid Contents

Considering the majority of the studies use colorimetric methods for the determination of the biological properties of tea TPC and TFC, the authors adopted the same methodology, allowing a better discussion of the results.

The TPC and TFC of black tea extracts were determined following the methodology described in Section 2.3. The TPC was determined by the Folin–Ciocalteu method described by Waterhouse [33]. An aliquot of 100 μL of each different tea extract at a concentration of 2 mg/mL was mixed with 100 μL of 2N Folin–Ciocalteu reagent plus 1500 μL of Milli-Q water, and then the mixture was vortexed for 20 s and placed in the dark for 3 min. Afterwards, 300 μL of 10% $Na_2CO_3$ (*w/v*) was added, homogenized, and then incubated at 50 °C for 5 min. The absorbance (Abs) of the samples was measured with a Shimadzu UV model 1800 at 760 nm against Milli-Q water. Gallic acid was used as a standard to produce a calibration curve at various concentrations. The results were expressed in milligrams of gallic acid equivalents per gram of dried extract (mg GAE/g of DE).

The main flavonols in tea are conjugates of quercetin and kaempferol varying from mono- to triglycosides plus lower levels of myricetin. The TFC was measured using the

colorimetric method of Chang et al. [34] with some modifications [30]. An aliquot of 100 μL of each different tea extract at a concentration of 2 mg/mL was mixed with 900 μL of Milli-Q water plus 100 μL of 10% $AlCl_3$ and 100 μL of 10% $KCH_3CO_2$. The mixture was then vortexed for 20 s and left at room temperature for 30 min. The absorbance (Abs) of the samples was measured at 415 nm, and rutin was used to produce a standard calibration curve at various concentrations. The results were expressed as mg of rutin equivalents per gram of dried extract (mg RE/g of DE).

### 2.6. Sample Preparation for Theaflavin Determination

The extraction of theaflavins from different tea samples was performed following the method published by Matsubara and Rodriguez-Amaya [35] with slight modifications. Briefly, 200 mg of each dry tea sample was brewed with a 20 mL solution (80% methanol in water) for 1 h 30 min at room temperature with mild stirring (250 rpm). The tea solution was filtrated through a 0.45 μm (pore size) cellulose acetate membrane to remove particulate matter. An aliquot of 10 mL of supernatant was concentrated in a rotary evaporator and reconstituted in 2 mL of methanol, and then 12.5 μL was submitted to HPLC analysis.

RP-HPLC Analysis of Theaflavins

For the theaflavin determination, a Hypersil ODS-5 μm (100 × 4.6 mm i.d.) column was used. Mobile phase A was composed of $H_2O$:formic acid (99.9:0.1, *v/v*), while mobile phase B was methanol:formic acid (99.9:0.1, *v/v*). To improve the selectivity of the separation, it was necessary to add a low formic acid concentration to suppress the ionization. The maximal response was achieved with 0.1% formic acid in the two mobile phases. The separation was achieved by a linear gradient in the following conditions: t = 0 min—30% B; t = 6 min—30% B; t = 40 min—39% B; t = 50 min—60% B; flow rate of 1 mL/min. The column, maintained at 40 °C, was attached to a 1260 Infinity II Quaternary Pump Liquid Chromatograph System, equipped with a PDAD fixed at 365 nm. The quantitative analyses were performed according to the external standard method using the OpenLab CDS VL Workstation software from Agilent Technologies (Avondale, PA, USA). To avoid retention time shifting and/or peak tailing, the sample concentration was restricted to the range of linearity that may occur when the sample amount approaches the column sample's load capacity. Peak identification was assigned by comparison with the authentic standards and further confirmed by superimposing the spectrum of each peak with the corresponding standard spectrum. The results were expressed as mg/g of the sample on a dry weight (DW) basis.

### 2.7. Extraction Methodology for Crude Catechin and Caffeine (CAF) Contents

The extraction of crude catechins and CAF was performed following the method published by Baptista et al. [36] with slight modifications. An amount of 100 mg tea extract, obtained by the methodology described in Section 2.3, was mixed with Milli-Q water in a 25 mL volumetric flask. An aliquot of 10 mL was partitioned in a separation funnel with an equal liquid volume of chloroform to remove pigments and other non-polar plant material. Then, 10 mL of ethyl acetate was added to the same volume of the aqueous solution for extraction. A similar extraction was repeated three times, and the three extracts were combined. Ethyl acetate was evaporated in a vacuum rotary evaporator, and the light-brown residue, called "crude catechins", was dissolved in 500 μL of water and then subjected to high-performance liquid chromatography/photodiode array detection (RP-HPLC/PDAD) analysis after being filtered through a 0.45 μm polytetrafluoroethylene membrane cartridge.

RP-HPLC Analysis of Catechins and Caffeine (CAF)

A relatively simple and quick RP-HPLC method was developed in which the catechins and CAF were simultaneously separated following the method of Baptista et al. [36], using a Phenomenex Synergi $C_{12}$—4 μm MAX-RP 80 A (150 × 4.6 mm i.d.) column. Mobile phase

A was composed of water:formic acid (99:1, *v/v*) (the addition of formic acid suppressed the ionization, improving the selectivity of the separation), while mobile phase B was acetonitrile. The separation was achieved by a linear gradient in the following conditions: t = 0 min—4% B; t = 60 min—25% B; flow rate of 0.7 mL/min. The column, maintained at 40 °C, was attached to a Waters HPLC (Waters 600 controller, 626 quaternary pump, and Waters 486 tunable absorbance detector fixed at 280 nm) system. The quantitative determinations were performed using a Hewlett-Packard integrator model HP-3396 series II. An aliquot of 10 μL injection volume was used, and the total run time was approximately 40 min. The chromatograms were recorded according to the retention times (RTs), and the quantitative analysis was achieved by the external standard method. As referred to in the description of the theaflavin determination, the catechin sample concentration was limited to the range of linearity. Peak identity was assigned based on the RT following the comparison with the authentic standards and/or by spiking the sample with the same standards, as already referred to in the description of the theaflavin determination. The average of triplicate measurements was used to calculate the catechin and CAF contents, and the results were expressed as milligrams per gram of sample on a dried extract basis.

### 2.8. Statistical Analysis

All determinations were performed, and the results are expressed as the means ± standard deviations (SD) of three independent measurements. One-way analysis of variance (ANOVA) was carried out to assess and indicate any significant differences between the mean values obtained from each sample. Significance was based on a confidence level of 95% ($p < 0.05$). Correlations between the tea quantity parameters and withering time evaluated were determined using Pearson's correlation coefficient (r). The statistical analysis was performed using SPSS 20.0 (SPSS Inc., Chicago, IL, USA).

### 3. Results and Discussion

#### 3.1. Effect of Withering Time on Antioxidant Activity of Camellia sinensis Black Tea

3.1.1. DPPH Free Radical Scavenging Activity (FRSA)

The 2,2-diphenyl-1-picrylhydrazyl (DPPH) assay was used to determine the antioxidant activity, and the results are presented in Table 1. The results are expressed as $EC_{50}$ values (μg/mL) (a lower $EC_{50}$ value is indicative of higher antioxidant activity). The free radical scavenging activity of black tea presents the best value at 9 h of withering (14.41 ± 0.27 μg/mL), followed by 16 h (14.50 ± 0.35 μg/mL), 12 h (14.88 ± 0.77 μg/mL), 6 h (14.94 ± 0.39 μg/mL), and, finally, 20 h (20.98 ± 0.25 μg/mL) of the withering process. Within the withering range of 6 to 16 h, the results do not show a significant difference ($p < 0.05$); however, the antioxidant activity decreases at 20 h. Based on the antioxidant activity decline caused by withering treatments, the smallest loss was observed with a withering time of 9 h. According to Ntezimana et al. [37], the smallest loss was observed at 8 and 10 h (29.48 ± 0.18 μg/mL and 30.32 ± 0.32 μg/mL, respectively), while the greatest was shown with 12 h of withering (32.29 ± 0.65 μg/mL). However, our study presents better results for free radical scavenging activity in comparison to those reported in alternative studies [37–39].

**Table 1.** Free radical scavenging activity (FRSA), ferric reducing antioxidant power (FRAP), ferrous-ion-chelating (FIC) activity, and extraction yield in dry extracts of Azorean black tea (*Camellia sinensis* var. *sinensis*) samples from the Gorreana Tea Plantation with different withering times.

| | FRSA (EC$_{50}$—µg/mL) [1] | FRAP (EC$_{50}$—µg/mL) [2] | FIC (%) | Extraction Yield (%) |
|---|---|---|---|---|
| 6 h | 14.94 ± 0.39 [a] | 15.73 ± 0.86 [c] | 36.40 ± 2.69 [d] | 26.73 ± 0.45 [a] |
| 9 h | 14.41 ± 0.27 [a] | 15.34 ± 0.26 [c] | 53.30 ± 2.69 [b] | 25.47 ± 0.35 [c] |
| 12 h | 14.88 ± 0.77 [a] | 15.54 ± 0.24 [c] | 46.96 ± 2.16 [c] | 26.13 ± 0.15 [b] |
| 16 h | 14.50 ± 0.35 [a] | 13.32 ± 0.31 [b] | 33.75 ± 2.47 [d] | 25.57 ± 0.38 [c] |
| 20 h | 20.98 ± 0.25 [b] | 22.27 ± 0.31 [d] | 55.36 ± 1.68 [b] | 25.73 ± 0.40 [bc] |
| BHT * | 44.89 ± 1.44 [c] | 5.99 ± 0.23 [a] | - | - |
| EDTA * | - | - | 97.64 ± 0.58 [a] | - |

Values are mean ± SD (*n* = 3). Different superscript letters indicate that values are significantly different (*p* < 0.05). [1] Half-maximal effective concentration. [2] Effective concentration at which the absorbance is 0.5. * At the same concentration of the extracts. BHT—butylated hydroxytoluene; EDTA—ethylenediaminetetraacetic disodium salt.

### 3.1.2. Ferric Reducing Antioxidant Power (FRAP)

Table 1 illustrates the FRAP results related to different times of withering for black tea samples, expressed as EC$_{50}$ values (µg/mL). The best results were found with 16 h of withering, exhibiting lower EC$_{50}$ values of 13.32 ± 0.31 µg/mL. However, between 6 and 12 h, the results do not show a significant difference (*p* < 0.05). The greatest loss of antioxidant decline was observed at 20 h of withering (22.27 ± 0.31 µg/mL). Our results present better values than those of other authors who presented an EC$_{50}$ value of 125 µg/mL [39].

### 3.1.3. Ferrous-Ion-Chelating (FIC) Activity

Table 1 shows the FIC activity after different withering times for black tea samples. The highest values were exhibited with 20 h of withering (55.36 ± 1.68%), followed by 9 h (53.30 ± 2.69%), and the lowest values were found for 16 h of withering (33.75 ± 2.47%). However, they were lower than the FIC activity of EDTA (97.64 ± 0.58%), a potent metal-ion chelator.

### 3.2. Total Phenolic Content (TPC) and Total Flavonoid Content (TFC)

Figure 1 illustrates the TPC and TFC values of black tea samples obtained for each different withering time. The TPC levels of the tea samples can be considered an indirect measure of their antioxidant activity because the basic redox mechanism of the Folin–Ciocalteu method was selected to screen phenolic content. The TPC results, expressed in milligrams of GAE/g DE, showed a slightly higher value at 9 h of withering (214.81 mg GAE/g DE), followed by 16 h (213.03 mg GAE/g DE), and the lowest value at 20 h (180.64 mg GAE/g DE). According to the study by Tong et al. [39], black tea samples presented lower values (102.98 mg GAE/g) compared to those reported in our study. However, Rohadi et al. [40] observed higher values (256.7 mg GAE/g) while Rahman et al. [41] also published slightly higher values for black tea samples in comparison to those in our study. These results can be explained by differences in climatic conditions, which in the Azores Islands are very unstable, particularly very windy; by the volcanic soil; by different processing conditions; and, particularly, by differences in genetics of the local *C. sinensis* plant. Another explanation for the fact that TPC values are very similar between 9 h and 16 h, and then decrease when withering exceeds 16 h, may be the fact that the physical withering takes place during the first 4 to 6 h (results also observed by Tomlins and Mashingaidze [42]), inducing the loss of moisture and changes in the permeability of the cell membrane which promote the softness of the tea leaf. However, after 16 h, the chemical withering is more intense, causing changes inside the leaf and consequently affecting the TPC contents to a greater extent.

Regarding TFC levels of black tea samples, determined by an aluminum chloride colorimetric method and expressed in milligrams of RE/g DE, differences between the different withering times were observed. The highest values were shown at 9 h of withering (72.60 ± 2.71 mg of RE/g DE), followed by 12 h (70.07 ± 2.77 mg of RE/g DE), and the lowest value was observed with 20 h of withering (59.73 ± 1.55 mg of RE/g DE). These results are correlated with the TPC values. However, these values are higher than those reported by Tong et al. [39], lower than those published by Rohadi et al. [40], and similar to those reported by Rahman et al. [41] for black tea samples.

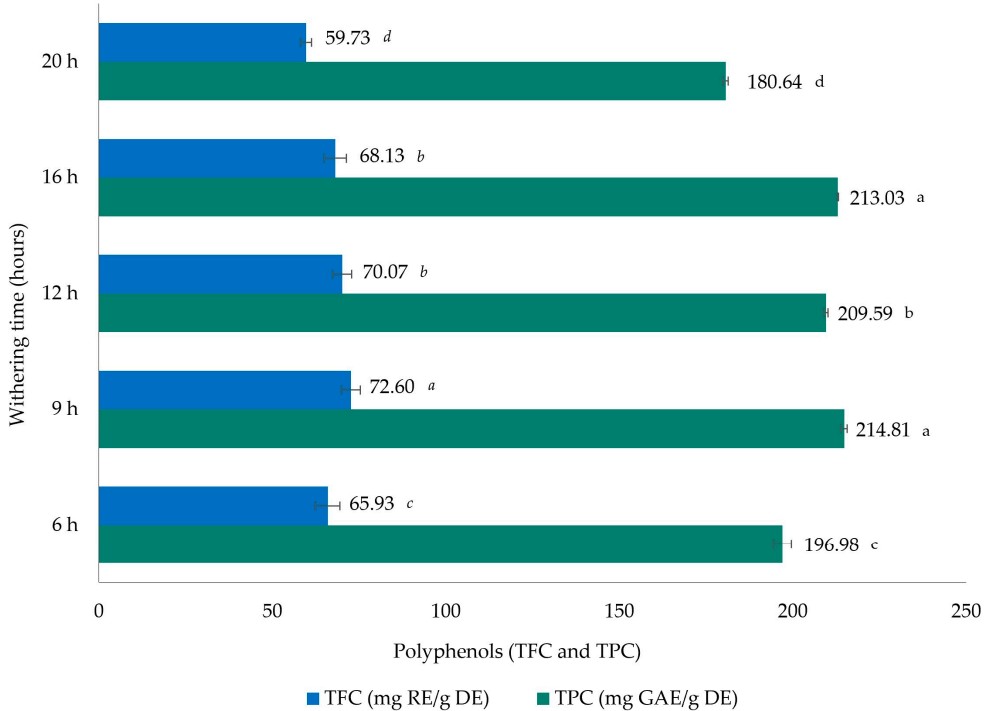

**Figure 1.** Total flavonoid content (TFC) and phenolic content (TPC) in dry extracts (DEs) of Azorean black tea (*Camellia sinensis* var. *sinensis*) samples from the Gorreana Tea Plantation with different withering times. Values are mean ± SD (*n* = 3). Different letters indicate that values are significantly different (*p* < 0.05). GAE—gallic acid equivalents; RE—rutin equivalents.

### 3.3. Determination of Theaflavin Content Profiles

It is well known that theaflavin levels are related to the taste and quality of black tea [13]. In this study, all theaflavin compounds (Figure 2) were well separated in a total run time of 60 min. Although most of the reported studies make use of a single-gradient long run with incomplete resolution, particularly regarding the minor components that are masked by the major nearby components, the authors decided to use two different 60 min gradients for the separation of catechins and theaflavins. A standard mixture of four theaflavins was eluted in the following order: TF, TF-3-G, TF-3′-G, and TF-3,3′-DG. In black tea leaves, the higher concentrations of theaflavin components are most likely a result of the oxidative polymerization of catechins, where most are converted into theaflavins [43]. TF-3,3′-DG is the galloylated theaflavin that has been shown to have the highest inhibition activity against SARS-CoV-2 3CLpro [44], consequently resulting in a lower degree of SARS-CoV-2 infectivity. This fact has stimulated, in the last three years, the scientific community to focus more attention on this aspect.

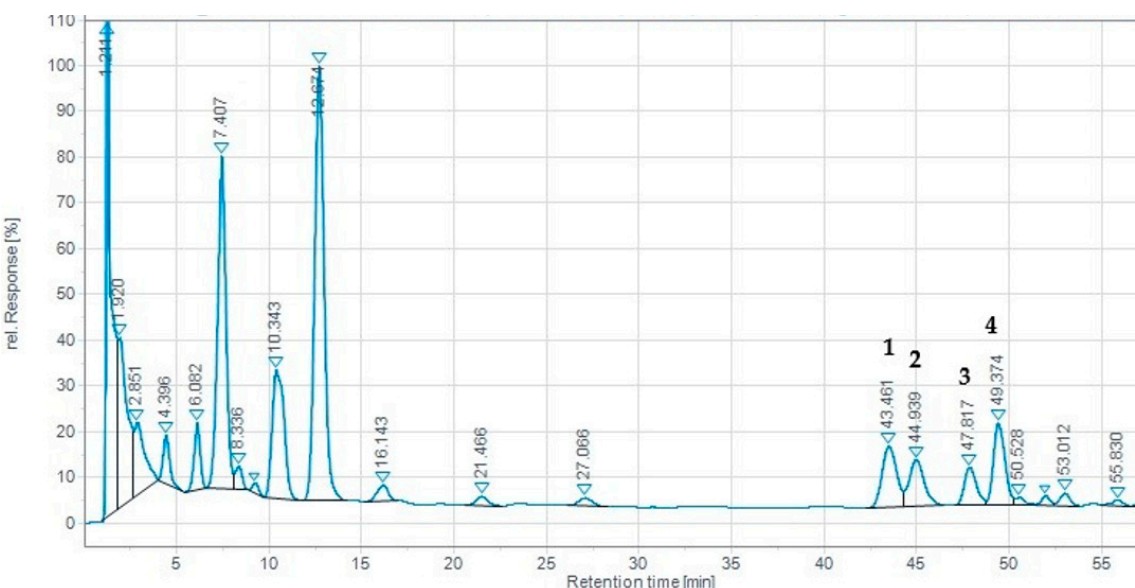

**Figure 2.** Reverse-phase HPLC chromatogram (UV at 365 nm) of Azorean black tea (*Camellia sinensis* var. *sinensis*) samples from the Gorreana Tea Plantation. Peaks: **1**. TF, theaflavin; **2**. TF-3-G, theaflavin-3-*O*-gallate; **3**. TF-3′-G, theaflavin-3′-*O*-gallate; **4**. TF-3,3′-DG, theaflavin-3,3′-di-*O*-gallate. Analytical chromatographic conditions are referred to in Section 2.

Figure 3 presents the theaflavin results obtained from black tea extracts with different withering times. According to our results, TF-3,3′-DG presents the highest value for 12 h (4.97 ± 0.03 mg/g DW) of withering time, followed by 9 h (4.29 ± 0.05 mg/g DW), while the lowest values were observed for 16 h (2.73 ± 0.01 mg/g DW). With respect to TF, the highest value was observed at 12 h (4.95 ± 0.03 mg/g DW); for TF-3-G and TF-3′-G, the highest values were found after 6 h of withering (5.25 ± 0.01 and 5.47 ± 0.01 mg/g DW, respectively).

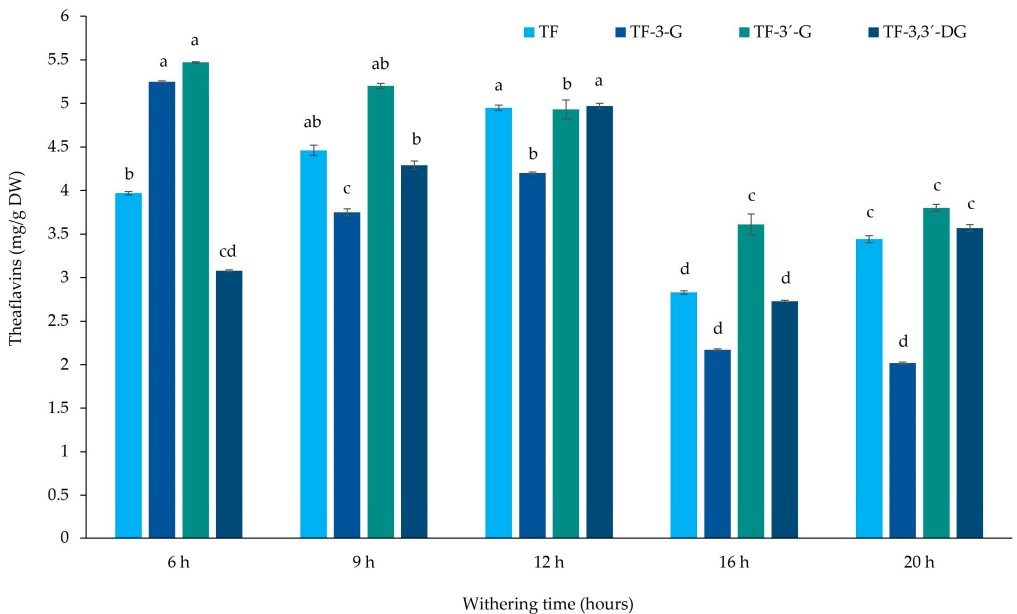

**Figure 3.** Theaflavin contents of Azorean black tea (*Camellia sinensis* var. *sinensis*) samples from the Gorreana Tea Plantation with different withering times (mg/g dry weight). Values are mean ± SD (*n* = 3). Different letters indicate that values between different withering times for each theaflavin are significantly different (*p* < 0.05). TF, theaflavin; TF-3-G, theaflavin-3-*O*-gallate; TF-3′-G, theaflavin-3′-*O*-gallate; TF-3,3′-DG, theaflavin-3,3′-di-*O*-gallate.

Figure 4 presents the total theaflavins. The highest contents were obtained at 12 h of withering (19.05 ± 0.39 mg/g DW), followed by 6 and 9 h (17.77 ± 0.83 and 17.70 ± 0.56 mg/g DW, respectively), while the lowest values were obtained at 16 and 20 h (11.34 ± 0.74 and 12.83 ± 0.48 mg/g DW, respectively). These results are higher than those reported by Macheka et al. [45], who presented a total theaflavin content of 3.24 mg/g following 16 h of withering. Both Lee et al. [46] and Rahman et al. [41] also presented a lower value for total theaflavins (6.37 and 6 mg/g DW, respectively) in comparison to our results. However, Zhang et al. [47] presented similar results for all theaflavins compared to our tea samples. Qu et al. [38] also presented similar results for TF-3,3′-DG but lower values for total theaflavins. Tong et al. [39] showed a much lower value for TF-3,3′-DG (0.27 mg/g), while Ramdani et al. [48] presented slightly higher values for TF-3,3′-DG (6.98 mg/g) and similar results for the total theaflavins (16.7 mg/g) in comparison to the values obtained in our study.

The slightly higher values of total theaflavins at 12 h of withering can be explained by the highest values observed for catechins at same time, and we observed a strong correlation with total theaflavins and catechins (r = 0.547), and more specifically with the sum of GC + GCG + C + CG (r = 0.651), and a moderate correlation with the sum of EGC + EC + EGCG + ECG (r = 0.470). Therefore, the theaflavin variations in tea leaves can be due to the origin, the location, the climatic conditions, the volcanic soil in the Azores Islands, and particularly the genetic differences of local tea plants. The theaflavins are responsible for the quality of black tea, and withering time should be limited to 18 h [24]. According to Owuor and Orchard [25], withering time beyond 20 h leads to the deterioration of black tea quality.

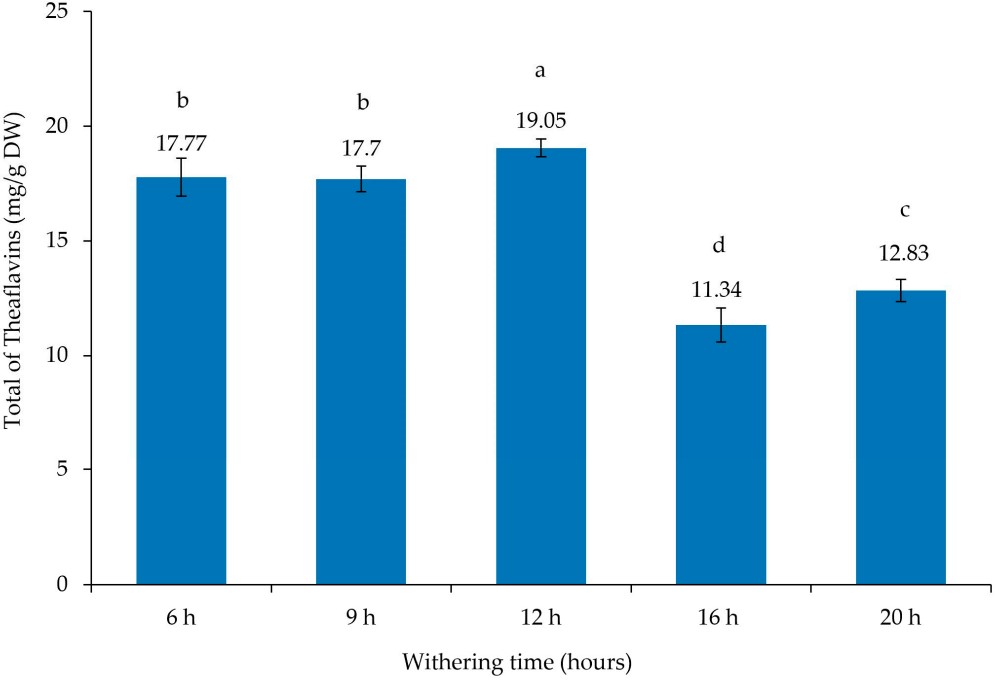

**Figure 4.** Total theaflavin contents of Azorean black tea (*Camellia sinensis* var. *sinensis*) samples from the Gorreana Tea Plantation with different withering times (mg/g dry weight). Values are mean ± SD (*n* = 3). Different letters indicate that values are significantly different (*p* < 0.05).

### 3.4. Determination of Catechin Content Profiles and Caffeine

The data presented in Table 2 presents the results related to the catechin and caffeine contents, according to different withering times, and Figure 5 shows the HPLC chromatogram of the catechin and caffeine profiles, the hydroxycinammate quinic acids, and theobromine. The hydroxycinammate quinic acids were identified by LC-MS (5-galloylquinic acid [M-H (*m/z* 343)], theobromine [M-H (*m/z* 181)], 3-caffeoylquinic acid,

[M-H (*m/z* 353)], 5-caffeoylquinic acid, [M-H (*m/z* 353)], and *p*-coumaroyquinic acid [M-H (*m/z* 337)]). The total catechin content was highest for 12 h (19.19 mg/g DW) of withering time, followed by 16 h (10.32 mg/g DW), and the lowest value was observed at 20 h (7.13 mg/g DW). For EGCG, the highest value was observed at 12 h (5.06 ± 0.12 mg/g DW), and the lowest value was observed at 9 h (2.35 ± 0.10 mg/g DW). Our results revealed that GC presented higher values, with the highest value observed for 12 h (8.00 ± 0.13 mg/g DW) and the lowest for 20 h (1.75 ± 0.08 mg/g DW). The higher values for GC in Azorean black tea can possibly be explained by genetic variation and differences in the environmental conditions, such as pedoclimatic factors and/or agricultural practices, or a combination of these factors.

**Table 2.** Catechin content (mg/g of catechins on a dry weight basis) and caffeine content in different samples of Azorean black tea (*Camellia sinensis* var. *sinensis*) samples from the Gorreana Tea Plantation with different withering times.

| Compounds (mg/g DW) | Withering Time (Hours) | | | | |
|---|---|---|---|---|---|
| | 6 h | 9 h | 12 h | 16 h | 20 h |
| CAF | 72.41 ± 4.11 [b] | 63.31 ± 2.61 [c] | 103.52 ± 5.46 [a] | 65.57 ± 0.77 [c] | 61.31 ± 0.77 [c] |
| GC | 4.16 ± 0.08 [b] | 3.18 ± 0.05 [c] | 8.00 ± 0.13 [a] | 3.87 ± 0.07 [bc] | 1.75 ± 0.08 [d] |
| EGC | 1.24 ± 0.16 [bc] | 1.33 ± 1.05 [b] | 1.84 ± 0.08 [a] | 1.21 ± 0.03 [c] | 0.82 ± 0.04 [d] |
| C | 0.15 ± 0.04 [c] | 0.27 ± 0.03 [b] | 0.50 ± 0.06 [a] | 0.31 ± 0.02 [b] | 0.12 ± 0.01 [c] |
| EC | 0.30 ± 0.07 [c] | 0.28 ± 0.07 [c] | 0.77 ± 0.02 [a] | 0.39 ± 0.02 [b] | 0.31 ± 0.03 [c] |
| EGCG | 2.59 ± 0.18 [c] | 2.35 ± 0.14 [c] | 5.06 ± 0.12 [a] | 3.05 ± 0.10 [b] | 2.96 ± 0.11 [b] |
| GCG | 0.56 ± 0.01 [b] | 0.29 ± 0.07 [d] | 0.80 ± 0.03 [a] | 0.47 ± 0.06 [c] | 0.30 ± 0.09 [d] |
| ECG | 0.79 ± 0.05 [b] | 0.67 ± 0.09 [c] | 1.49 ± 0.10 [a] | 0.66 ± 0.07 [c] | 0.62 ± 0.02 [c] |
| CG | 0.45 ± 0.02 [b] | 0.28 ± 0.02 [cd] | 0.73 ± 0.04 [a] | 0.36 ± 0.06 [bc] | 0.25 ± 0.04 [d] |
| ECDs | 4.92 [c] | 4.63 [c] | 9.16 [a] | 5.31 [b] | 4.71 [c] |
| Total Catechins | 10.24 [b] | 8.65 [c] | 19.19 [a] | 10.32 [b] | 7.13 [d] |

Values are mean ± SD (*n* = 3). Different superscript letters indicate that values between different withering times for each individual catechin and caffeine are significantly different (*p* < 0.05). DW—dry weight. CAF—caffeine; GC—gallocatechin; EGC—epigallocatechin; C—catechin; EC—epicatechin; EGCG—epigallocatechin-3-*O*-gallate; GCG—gallocatechin-3-gallate; ECG—epicatechin-3-gallate; CG—catechin-gallate; ECDs (epicatechin derivatives)—sum of EC, EGC, EGCG, and ECG.

For EGC, EC, ECG, GCG, and CG, the highest values were observed for 12 h (1.84, 6.13, 1.49, 0.8, and 0.73 mg/g DW, respectively). Catechin alone presented lower values in the Azorean *C. sinensis* in comparison to other individual catechins and the highest value was observed at 12 h, and the lowest value was observed at 20 h (0.5 and 0.12 mg/g DW, respectively). For the total ECDs, the highest value was observed at 12 h (9.16 mg/g DW), and the lowest value was observed at 9 h (4.63 mg/g DW). Relative to CAF, the results were higher and very similar among 6, 9, 16, and 20 h of withering time (72.41, 63.31, 65.57, and 61.31 mg/g DW, respectively); nevertheless, a major difference was observed for 12 h (103.52 mg/g DW) of withering. According to Owuor and Orchard [25], an increase in caffeine content in tea appears to be related to the breakdown of proteins into amino acids and their metabolism. Similar results have also been observed by other authors [48,49], both for individual and total catechins; however, for caffeine, the results were different, with higher values observed in Azorean tea samples.

The highest values of individual catechins and total catechins at 12 h of withering can be explained by the highest values observed for the sum of catechins in the same time range. According to Table 3, the highest values of total catechins can be explained by the strong correlations between the total catechins and the sum of GC + GCG + C + CG (r = 0.976), between the total catechins and the sum of EGC + EC + EGCG + ECG (r = 0.980), and between the sum of GC + GCG + C + CG and the sum of EGC + EC + EGCG + ECG (r = 0.923), indicating that for all withering times, the catechins are strongly correlated.

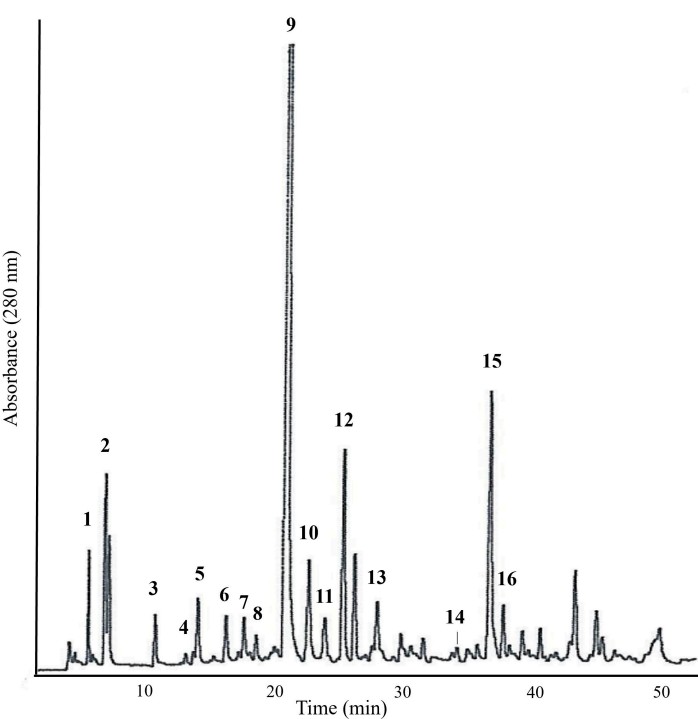

**Figure 5.** Reverse-phase HPLC chromatogram (UV at 280 nm) of Azorean black tea (*Camellia sinensis* var. *sinensis*) extract from the Gorreana Tea Plantation. Identified peaks: **1**. GA—gallic acid; **2**. 5-galloylquinic acid; **3**. GC—gallocatechin; **4**. theobromine; **5**. 3-caffeoylquinic acid; **6**. EGC—epigallocatechin; **7**. C—catechin; **8**. 5-caffeoylquinic acid; **9**. caffeine; **10**. unknown; **11**. EC—epicatechin; **12**. EGCG—epigallocatechin-3-O-gallate; **13**. p-coumaroyquinic acid; **14**. GCG—gallocatechin-3-gallate; **15**. ECG—epicatechin-3-gallate; **16**. CG—catechin-gallate. Analytical chromatographic conditions are referred to in Section 2.

**Table 3.** Correlation matrix of the studied parameters in dry extracts of Azorean black tea (*Camellia sinensis* var. *sinensis*) samples with different withering times (Pearson correlation coefficients).

|  | TPC | TFC | Total Catechins | Total TFs | GC + GCG + C + CG | EGC + EC + EGCG + ECG |
|---|---|---|---|---|---|---|
| TPC | 1 | - | - | - | - | - |
| TFC | 0.952 | 1 | - | - | - | - |
| Total catechins | 0.233 | 0.306 | 1 | - | - | - |
| Total TFs | −0.196 | −0.358 | 0.547 | 1 | - | - |
| GC + GCG + C + CG | 0.258 | 0.278 | 0.976 | 0.651 | 1 | - |
| EGC + EC + EGCG + ECG | 0.280 | 0.369 | 0.980 | 0.470 | 0.923 | 1 |

TPC-total phenolic content. TFC-total flavonoid content. TFs-theaflavins.

*3.5. Pearson Correlations between Parameters*

Pearson correlation was used to assess the relationship between total phenolic, flavonoid, catechin, and theaflavin contents.

Some significant correlations were observed among the methods used to determine the biological activities (Table 3). TPC and TFC were strongly correlated (r = 0.952), and the same patterns were observed for total catechins and the sum of GC + GCG + C + CG (r = 0.976), and the sum of EGC + EC + EGCG + ECG (r = 0.980), and between the sum of GC + GCG + C + CG and the sum of EGC + EC + EGCG + ECG (r = 0.923). Between total TFs and the sum of GC + GCG + C + CG (r = 0.651) and between total TFs and total catechins (r = 0.547) we also observe a strong correlation.

A moderate correlation was observed between total TFs and the sum of EGC + EC + EGCG + ECG (r = 0.470), between TFC and EGC + EC + EGCG + ECG (r = 0.369), and between TFC and total catechins (r = 0.306).

A weak correlation was observed between total catechins and TPC (r = 0.233), between TPC and GC + GCG + C + CG (r = 0.258), between TPC and EGC + EC + EGCG + ECG (r = 0.280), and between TFC and GC + GCG + C + CG (r = 0.278), while a weak and negative correlation was observed between total TFs and TPC (r = −0.196), and a moderate and negative correlation was observed between TFs and TFC (r = −0.358).

## 4. Conclusions

These results clearly highlight differences in the bioactivity and quality of Azorean black tea according to different withering process conditions. The FRSA, FRAP, and FIC activities presented different values according to the different withering conditions, showing, for FRSA, similar results within the withering range of 6 to 16 h, with the highest value being observed at 9 h and the lowest value for 20 h. For FRAP, the best results were observed at 16 h, and between 6 and 12 h, no significant differences were observed. Conversely, for FIC, the highest value was observed at 20 h, and the lowest value was observed at 16 h.

The TPC and TFC values were correlated, showing the highest value at 9 h and the lowest at 20 h.

For the total theaflavins, the highest values were obtained at 12 h, followed by 6 and 9 h of withering, and the lowest value was obtained at 20 h.

According to different withering times, the total catechins and ECDs were higher at 12 h, followed by 16 h, and the lowest value was observed for 20 h. Linked to this, some differences in the catechin contents are due to genetic variation, and others may be related to environmental conditions, such as pedoclimatic factors and/or agricultural practices, or a combination of these two factors. Relative to CAF, the results were very similar between 6 and 20 h of withering, with the exception of 12 h, which presented the highest value.

In conclusion, in all studied parameters, the best withering times were observed within the range of 9 to 16 h, showing decreased values at 20 h, with the exception of FIC, which may possibly be due to the chemical degradation of tea components after 20 h.

**Author Contributions:** L.S.P., A.P.D., M.F.M. and J.A.B.B. contributed to the design and implementation of the research, to the analysis of the results, and to the writing of the manuscript. All authors have read and agreed to the published version of the manuscript.

**Funding:** This work was financially supported by Secretaria Regional da Agricultura e do Desenvolvimento Rural—Direção Regional da Agricultura (No. AD/2023/7/DRAG).

**Data Availability Statement:** All data are contained within the article.

**Acknowledgments:** The authors (L.S.P., A.P.D., M.F.M., and J.A.B.B.) are thankful to Gorreana Tea Plantation for the tea samples used in this study.

**Conflicts of Interest:** The authors declare no conflict of interest.

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
