# Peer review of "The Impact of Different Withering Process Conditions on the Bioactivity and Quality of Black Tea from Azorean Camellia sinensis"

_beverages, doi:10.3390/beverages9040094_

Round 1

Reviewer 1 Report

Comments and Suggestions for Authors

The study investigated the effect of withering times on the content of black tea bioactive compounds (total polyphenols and flavonoids, TFs, CAF) and its antioxidant activity in vitro. 

Camellia sinesis is one of the best-studied crop plants. The profile of tea bioactive compounds is well known, as is its health-promoting potential. Despite this, researchers are still interested in this species, which is also confirmed by the reviewed publication.

In my opinion, the scientific quality of the article is rather average. The Authors generally continue the methodological concept from their previous work, published in LWT and Curr. Res. Food Science.

The clarity of the presentation does not raise any major objections to me.

However, I believe that both total polyphenolic compounds and flavonoids should be determined using the RP-HPLC technique. The Authors use it in their research to determine TFs and CAFs. For example, the Folin method is burdened with low specificity.

Why did the Authors decide to use spectrophotometry? Is it possible to present the results of chromatographic tests? I do not require full identification of individual compounds, e.g., from the group of flavan-3-ols, polymerized proanthocyanidins, phenolic acids, or flavonols, but only their total content. Remodeling the methodology would allow to develop and substantively improve the discussion of the results.

I will be grateful for your response to my comment.

Reviewer 2 Report

Comments and Suggestions for Authors

The paper of Paiva et al. “The impact of different withering process conditions on the bioactivity and quality of black tea from Azorean Camellia sinensis” aimed to study of biochemical changes if withered Azorean Camellia sinensis. Generally, the paper contains flaws in methodology reduced those small advantages that could be estimated as positive.

Highlights and strengths of the manuscript are:

The results may further increase interest in Azorean Camellia sinensis as a natural beverage and help develop new strategies for its application.  

Specific comments and suggested revisions:

Introduction. In the absence of consensual clarification of the necessity to study influence of withering times on the biochemical parameters of Azorean black tea, the importance of the study was little evident. The paper Ntezimana et al. (Different Withering Times Affect Sensory Qualities, Chemical Components, and Nutritional Characteristics of Black Tea. Foods 2021, 10, 2627. https://doi.org/10.3390/foods10112627) and many other were already demonstrated similar results. Maybe it makes sense because the Azorean black tea was not study before, but in this case, the idea needs to be clarified. It looks like another example of known experiment.

Withering times. Why 6, 9, 12, 16, and 20 hours were the basic points, but 0 and 3 hours were excluded? Was it done using a known method or methodology, or why?

Total Flavonoid Content. Because of deficiencies in the flavonoid analysis assay, the paradox of flavonoid accumulation in cut plants was observed – flavonoid level after 6 h was 65.93 and after 9 h was 72.60 (unit of measurement is unknown). The dead plant is not capable to produce flavonoids. Disadvantages of colorimetric method could be avoided by application of HPLC.

Extraction Methodology for Theaflavins. It is not clear the principle of extraction methodology for theaflavins. It is just 80% ethanol extraction of tea material without any purification (or separation steps) used for the concentration of target compounds. It is not clear why chloroform and ethyl acetate extraction of 80% ethanol extract (used for theaflavin analysis) used to obtain catechin (+ caffeine) concentrate. Why can't you take the 80% ethanol extract and separate it in HPLC assay. Theaflavins characterized higher retention times comparing with catechins so they don't interfere (which is clearly visible in your figure 2).

Theaflavin content. Due to the absence of 0-hour data it appears that original tea material characterized by the non-zero level of theaflavins what is wrong. Sometimes one gets the impression that the theaflavin‐3/3’‐O‐gallates content reduced during withering, which makes no sense.

Caffeine content. The same claims as in case of flavonoid content. How would you explain the accumulation of caffeine in cut plants?

With all due respect to authors, I see no possibility to recommend paper for publication in present form. The paper has serious flaws that can be corrected by adding new data and additional experiments.

Round 2

Reviewer 2 Report

Comments and Suggestions for Authors

The authors taking into account the observations made by the reviewer. Considering the explanation provided by the authors the paper after correction may be accepted in present from.

Author Response

We would like to express our deepest appreciation to the reviewer for their comments about the manuscript.